# Experimental Study on Mechanical Properties of Coal-Based Solid Waste Nanocomposite Fiber Cementitious Backfill Material

**DOI:** 10.3390/ma16155314

**Published:** 2023-07-28

**Authors:** Qiangqiang Cheng, Haodong Wang, Yaben Guo, Bin Du, Qixiang Yin, Linglei Zhang, Yue Yao, Nan Zhou

**Affiliations:** 1School of Architecture and Construction, Jiangsu Vocational Institute of Architectural Technology, Xuzhou 221000, China; chengqiang_205@163.com (Q.C.); dubin_china@163.com (B.D.); yinqixiang1988@163.com (Q.Y.); zlinglei@163.com (L.Z.); cm.522.1314@foxmail.com (Y.Y.); 2School of Mining Engineering, China University of Mining and Technology, Xuzhou 221000, China; guoyaben819@163.com (Y.G.);

**Keywords:** coal-based solid waste disposal issues, nanocomposite fibers, nano-SiO_2_ doping, curing ages, mechanical performance test

## Abstract

Previous studies have shown that coal-based solid waste can be utilized in combination with cement, silica fume, and other modified materials to create a cemented backfill material. However, traditional cemented backfill materials have poor mechanical properties, which may induce the emergence of mining pressure and trigger dynamic disaster under complex mining conditions. In this study, the nanocomposite fiber was used to modify the traditional cemented backfill materials and a new cemented backfill material was developed using coal-based solid waste, nanocomposite fiber and other materials. Specifically, coal gangue, fly ash, cement, and glass fibers were used as the basic materials, different mass fractions of nano-SiO_2_ were used to prepare cemented backfill materials, and the mechanical enhancement effect of the compressive strength, tensile strength, and shear strength of the modified materials was analyzed. The results show that when the nano-SiO_2_ dosage is 1%, the optimal compressive strength of the specimens at the curing age of 7 d can be obtained compared with cemented materials without nano-SiO_2_, and the compressive strength of the modified specimens raises by 84%; when the nano-SiO_2_ dosage is 1%, the optimal tensile strength and shear strengths of the modified specimens can be obtained at the curing age of 28 d, increasing by 82% and 142%. The results reveal that nanocomposite fibers can be used as additives to change the mechanical properties of cemented backfill materials made using coal-based solid waste. This study provides a reference for the disposal of coal-based solid waste and the enhancement of the mechanical properties of cemented backfill materials.

## 1. Introduction

Coal constitutes a significant component of the global energy industry, with an estimated production exceeding 7 billion tons in 2018 alone [1]. However, the process of coal mining results in the generation of a significant amount of mine waste, which can lead to environmental pollution [2]. Coal-based solid waste refers to industrial solid waste generated during coal mining, which mainly includes coal gangue and fly ash [3]. Generally, these solid wastes are dumped into the ground, resulting in land occupation, surface subsidence, soil erosion, and ecological damage [4]. At present, the comprehensive disposal rate of coal-based solid waste in many countries is less than 60% [5]. The disposal of coal-based solid waste is dominated by surface piles and underground landfills. The existing studies have shown that the discharge of coal-based solid waste not only wastes land resources and causes environmental pollution and ecological damage, but also induces geological disasters in a mining area [6].

The extensive accumulation of coal-based solid waste is a prevalent concern in coal mines around the world; therefore, the disposal of coal-based solid waste has become an important factor affecting resource development and environmental protection, and the comprehensive utilization of coal-based solid waste on a large scale is an important approach to address the issue of solid waste treatment. The utilization of coal-based solid waste was initially adopted in developed Western countries and garnered significant attention worldwide until the late 1960s. In the 1970s, the slogan of “Resource recycling” was put forward, and the United States, Germany, the Netherlands, and other countries realized a relatively high degree of utilization of coal gangue and formulated corresponding technical standards. In terms of the comprehensive use of coal gangue, an integrated treatment and utilization system has been formed in many countries for power generation, paving, the production of building materials, the production of chemical raw materials, agricultural applications, and underground filling. For example, coal gangue has been used to generate electricity in coal mine power plants and coal plants of Germany, the Netherlands, and other countries since the 1970s [4,7]. Coal gangue was also used as a road base or subgrade fill gangue in France, Germany, the United Kingdom, and other major motorway countries [8].

At present, coal-based solid waste made of cemented backfill material are widely used in mining engineering. The main composition of cemented backfill materials includes solid wastes (such as tailings, slag, river sand, coal gangue, and fly ash) with a mass proportion of about 70–85% as an aggregate, additives (such as cement), and water; after the preparation, cemented backfill materials are transported to the goaf to support the surrounding rock [9]. This backfilling method not only effectively controls the roof pressure disaster in a mining area but also plays an important role in the recovery of hazardous resources [10]. However, with the continuous updating of cemented backfill technology, there is an urgent need for the material to be both high quality and cost effective to achieve the optimal effect on roof pressure control in the mining area. According to statistics, the cost of cemented backfill materials accounts for 20% of the total mining cost, of which the cost of cement can account for about 75% of the cost of cemented materials [11]. In the field of civil engineering, cemented materials are mixed with other additives such as lime, fly ash, slag, or other alkaline catalysts [12,13], which can effectively reduce the cost and improve the performance of cemented materials. Some researchers have shown that incorporating the binder with different amounts of silica fume (0%, 5%, and 20% of the total mass of the binder) as an additive can enhance the microstructure and properties of the material, thus increasing its mechanical properties [14]. The incorporation of weak acids (0.5%, 1.5%, and 2.5%) into the composition of Magnesium Oxygen Sulphate (MOS) cement, a durable green building material, can significantly enhance the compressive strength and water resistance of MOS cement pastes [15]. The permeability of cemented materials is the main performance characteristic of their durability, in which a permeability-reducing agent (PRA) is incorporated to reduce the permeability and improve the durability of building materials [16]. Cheng Qiangqiang [17] used fly ash cement to modify solid sea-phase clay for material modification and analyzed its yield strength, stiffness properties and other properties. This study provides a theoretical foundation for the engineering application of fly ash hydromorphic clay. Concrete is a commonly used material in the construction industry. Relevant studies have shown that by incorporating silica fume, the strength, quality, and the depth of strength degradation of the modified concrete material can be significantly increased [18]. In addition, backfill materials can also be improved by material modification. For example, industrial waste can be mechanically activated in the crusher to replace conventional waste during the material preparation, so as to improve the rheology and mechanical strength of backfill composite materials [19]. Utilizing filling mixtures derived from crusher-activated rock salt waste not only reduces the cost of the filler but also improves the flow and mechanical properties of the material [20]. It is also found that the disintegration and activation of halloysite-enriched waste in a disintegrator followed by mixing with other fillers improves the lightness and rheological properties of the material, suggesting that the quality of the cementitious material can be improved by disintegration of the activated components [21]. In the mining field, the mechanical properties of cemented backfill materials are key to safe and efficient mining after the materials have been backfilled to the goaf [22]. However, the cemented backfill materials composed of tailings, cements, and other mixed solids [23] have poor mechanical properties and cannot provide good support for a coal face with complex stress distribution and strong pressure. In this regard, many researchers have tried to add other reinforcing materials to improve its mechanical properties [24].

Nanocomposite fibers are crystalline materials with high chemical activity [25], which can significantly enhance the mechanical properties of cemented materials such as tensile strength and elastic modulus [26,27]. Nanocomposite fiber material is a novel single-crystalline or polycrystalline substance with a diameter of 20 nm and a length of 1–100 nm, which exhibits superior properties such as high tensile strength and minimal environmental impact [28,29]. A small amount of carbon nanofibers has a more pronounced effect on the properties of the composites. When the mass of the nanocarbon fiber is 0.3%, the compressive strength, bending strength and splitting tensile strength of the modified material are increased by 9.2%, 13.2% and 17.5%, respectively [30]. Based on the modification of alumina nanocomposites, the incorporation of carbon fibers can significantly improve the tensile properties of the composites [31]; when water-soluble ore-enriched wastes are used as backfill materials, the incorporation of nano-siliconized additives result in a significant increase in the strength properties of the composite backfill materials and a reduction in the consumption of adhesives [32]. Nanocomposite fiber materials can be used to effectively reduce the thermal conductivity of sand slurry and increase the compressive strength of cement mortar [33]. Nano-SiO_2_ as an additive can effectively improve the mechanical properties and durability of ultralight cementitious composites [34], and cellulose filament (CF) is a novel nano-cellulose material that can effectively modify the rheological properties of composites [35]. Chen et al. [36] explored the compressive and tensile strength of fiber-modified materials. Liu Zhongkun et al. [37] analyzed the effects of different amounts of nano-SiO_2_ on the workability, microstructure, and compressive strength of concrete samples after early negative-temperature curing and concluded that this method can effectively improve the microscopic hydration structure and increase the compressive strength of the material. In some instances, the incorporation of chemical additives into nanofibrous materials can enhance the brightness of composites. Research has demonstrated that nano-modified suspensions containing thin hydrotalcite, graphene, and carbon nanotubes as additives can also improve concrete strength [38]. The incorporation of a combination of poly(vinyl alcohol) (PVA) fibers and metakaolin (NS) into fly ash mortar led to a significant enhancement in both mechanical performance and resistance against chloride penetration [39]. Some studies have also reported that the incorporation of nanoscale materials changes the microstructure of cemented materials, thus improving their mechanical properties. Alrekabi prepared a novel cementitious material incorporating the synergistic effect of nanocomposite fibers and microfibers and examined the microstructure of the material scanning electron microscope (SEM). The results demonstrated that the development of cracks in cemented materials was significantly reduced [40].

In summary, nanocomposite fiber materials can serve as additives to enhance the properties of cemented materials (such as cement, concrete, mortar, fly ash, etc.), leading to significant improvements in the macromechanical and other properties of the materials. However, the majority of studies have been conducted in the field of civil engineering, and there are quite a few studies on the modification of coal-based solid waste materials using nanocomposite fibers in the field of mining. To this end, nanocomposite fibers were employed to modify coal-based solid waste cemented backfill materials, and nanocomposite fiber materials were applied in the field of backfilling mining in this study. Besides, experiments on mechanical properties of cemented backfill materials made of coal-based solid waste and nanocomposite fiber were conducted, and the effects of different nano-SiO_2_ dosages and curing ages on the tensile strength, compressive strength and shear strength of the modified cemented backfill materials were analyzed. The enhancement degree of the mechanical properties of the modified cemented materials was discussed, and the optimal nano-SiO_2_ dosages and curing age were obtained. This study provides a theoretical basis for the preparation of cemented backfill materials made of nanocomposite fibers and coal-based solid wastes.

## 2. Methods and Materials

### 2.1. Test Materials and Equipment

#### 2.1.1. Test Equipment

The equipment required for this test included an electro-hydraulic servo press, a jaw crusher, an X-ray diffractometer, a standard curing box, and a hand-held mixer. Figure 1 shows pictures of the relevant equipment, and Table 1 shows the specific parameters of the equipment.

#### 2.1.2. Test Materials

According to the continuity of engineering practice and materials research [40,41], fly ash, cement, gangue, glass fibers, nano-SiO_2_, and water were used to prepare the cemented backfill materials, as shown in Figure 2.

Coal gangue

Coal gangue is one of the main solid wastes in the mine. In this test, irregular black-grey gangue taken from a mine in Shandong Province was selected. The gangue samples were crushed to less than 20 mm by a jaw crusher, the curves of particle size grading were drawn in Figure 3. The diffraction analysis of the gangue powder was conducted with the X-diffractometer. It was determined that the gangue was mainly composed of quartz, zeolites, and kaolin, as shown in Figure 4.

2.Fly ash

The fly ash used in this test was purchased. According to Chinese standard GB1596-91 [42], fly ash at level III was employed. The selected fly ash exhibited a greyish-white hue and predominantly existed in powder form, with only a minor proportion of agglomerates. Figure 5 shows the X-ray diffraction analysis of fly ash powder. The main components of fly ash are erythrite, quartz and mullite, of which quartz and mullite are the main components, while erythrite is present in trace amounts.

PO42.5 silicate cement was used, and the diffraction pattern of the cement was obtained via X-ray diffraction, as shown in Figure 6. The main components of this cement are calcium silicate, calcium carbonate, quartz, and gypsum.

3.Glass fibers with nano-SiO_2_

According to the continuity of the test [43], glass fibers and nano-SiO_2_ were purchased from companies in Shanghai and Suzhou, China. Table 2 and Table 3 show the specific parameters of glass fibers and nano-SiO_2_.

### 2.2. Experimental Process

The previous studies have shown that the best mechanical properties and conveying properties of cemented materials could be obtained when the mass proportion of gangue- and fly ash-based cementitious material was 81% [44], or that of sand-based cementitious material was 76% [45]. Based on the existing studies [43], the mass ratio of nanocomposite fibers cementitious material was determined to be 79%, in which fly ash, gangue, cement, and glass fibers accounted for 50%, 40%, 10%, and 0.3%, respectively. The experimental results have shown that the incorporation of nanometer additives at a concentration of 0.5% or lower into cemented materials results in enhanced mechanical properties, whereas higher dosages of additives result in inferior performance of cemented materials across all aspects [46,47]. In this test, cemented material specimens with a size of 70.7 mm × 70.7 mm × 70.7 mm were prepared, and different nano-SiO_2_ dosages of 0%, 0.1%, 0.5%, and 1% were added to specimens, respectively. Using the electro-hydraulic servo universal testing machine (WAW-1000D), uniaxial compressive, tensile and shear tests on specimens cured for 3 d, 7 d, 14 d, and 28 d were conducted at the shear angles of 40°, 50° and 60°, and the compressive strength, tensile strength, and shear strength of the specimens were analyzed [48]. A constant loading rate of 0.5 mm/s was employed until specimen failure occurred, and experimental data were recorded. Table 4 shows the specific test program.

### 2.3. Specimen Preparation

According to the above test program, due to the limitations of the multi-functional mixer and mixing bucket, the cemented material could only be stirred in a single group and in a single-component type. Each group was prepared with 3 specimens, and the average value of the three specimens was taken as the final experimental result. Fly ash, gangue, cement, and glass fibers required for a single group were added to the mixing bucket for dry mixing. After the ingredients were stirred evenly and full contact was achieved, nano-SiO_2_ was added for dry mixing and then tap water was added and mixed for 3–4 min. The mixing time can be extended appropriately to ensure the homogeneity of the additives. According to Standard for test method of performance on building mortar [49], the prepared slurry was poured into the standard mold with a bottom of 70.7 mm × 70.7 mm × 70.7 mm, a small amount of oil was evenly coated on the side and bottom of the mold, and the vibration table was activated after filling one-third of the mold volume. When the mold was completely filled with the slurry, the mold was left to stand still for a while. After the slurry was completely filled, the mold was kept stationary for 24 h and transferred to a maintenance box at a temperature of 20 ± 0.5 °C and a humidity of 95% RH. Figure 7 shows the details of the preparation process.

## 3. Results and Discussion

### 3.1. Compressive Strength Test

#### 3.1.1. Compressive Failure Mode

Taking the specimen with a nano-dosages of 0.5% as an example, the compressive damage morphology of cemented specimens made of coal-based solid waste and nanocomposite fibers is obtained, as shown in Figure 8.

As shown in Figure 8, it was found that:(1)Cracks developed from the corners of the specimen at the curing age of 3 d. With the increase in load, the cracks at the corners of the specimen gradually propagated upwards until they penetrated through the entire specimen. When the cracks in the corners of the specimen developed, cracks in the middle of the specimen also appeared, while they did not pass through the whole specimen. The side of the specimen exhibited two large cracks, and a large number of fibers were observed in the cracks, which pulled each other to maintain the complete form of the specimen.(2)The specimen at the curing age of 7 d was damaged from the left side of the lower corners. With the increase in the load, the two small cracks on the left side of the specimen developed and expanded, and eventually penetrated the whole specimen. The penetrating cracks were located in the middle of the specimen, and the width of the specimen reached the maximum. As the loading time increased, small cracks were generated on the right side of the specimen, and large cracks appeared in the middle of the specimen and extended to the end of the specimen.(3)The damage location of the specimen at the curing age of 14 d was the same as that of 7 d, and the damage on the specimen started from the left side of the lower corners. At the beginning, two cracks were generated in the corners of the specimen. With the increase in load, the two cracks developed upward, and the specimen was constantly broken in the process. Due to the fiber tension, the specimen still maintained high integrity, and the two cracks in the corners eventually penetrated through the whole specimen. Small cracks were also generated on the left side of the specimen.(4)The specimen at the curing age of 28 d was damaged from the right side of the lower corners. The cracks formed on the right side and in the middle of the specimen simultaneously. With the increase in load, two cracks developed and extended 1/2 of the specimen’s length, and finally cracks formed through the middle of the specimen at an angle of 45°. There was a peeling phenomenon on the surface of the right side of the specimen, while the specimen still maintained high integrity. Due to the development of cracks on the left surface of the specimen, a large number of fragments fell off, and the evenly distributed glass fibers can be seen on the cross-section of the fallen fragments.

This shows that the compressive failure morphology of specimens exhibits different forms at different curing ages. With the increasing age, the specimen changed from a small amount of debris peeling to a large amount of debris shedding, and the number of cracks also increased with the increasing curing age in the compressive test.

#### 3.1.2. The Effect of Different Nano-SiO_2_ Dosages on Compressive Strength

Figure 9 shows the stress–strain curves of compressive strength of cemented backfill materials with different nano-SiO_2_ dosages.

As shown in Figure 9, the compressive strength of the specimens containing 0.1% and 0.5% nano-SiO_2_ was not significantly improved in the early stage of curing. The peak compressive stress of the specimen containing 1% nano-SiO_2_ was 5.78 MPa at the curing age of 14 d. At the curing age of 28 d, the overall peak stress of the specimens became larger; the compressive strength of the specimens containing 0.5% and 1% nano-SiO_2_ reached about 6.7 MPa, and the compressive strength of the specimens containing 1% nano-SiO_2_ significantly improved. With the extension of the curing period, the compressive strength of the specimens increased significantly with the increase in nano-SiO_2_ dosage. However, as the curing age reached 28 d, the compressive strength of the specimens no longer significantly increased with the increase in nano-SiO_2_ dosage. The main reason is that when a small amount of nano-SiO_2_ was added, nano-SiO_2_ could not fully participate in the reaction of volcanic ash in the early curing stage, and less of the C-H-S gel produced by the cement hydration was generated [47], contributing to the lower compressive strength of the specimens than the bonded specimens only mixed with glass fiber. When a large amount of nano-SiO_2_ was added, nano-SiO_2_ fully participated in the reaction of volcanic ash, and the cement hydration reaction was promoted, resulting in a small difference in compressive strength among specimens with different nano-SiO_2_ contents after the curing age of 28 d. This finding was consistent with the previous results in the literature [46], indicating that the addition of excessive nano-SiO_2_ had a negative impact on the improvement of compressive strength.

Therefore, for the early strength of the specimen, an increase in the amount of nano-SiO_2_ had a significant impact on the compressive strength of the specimen. However, the addition of more nano-SiO_2_ dosage was unable to significantly enhance the later strength of the nanocomposite fiber bonded specimen.

Figure 10 shows the stress–strain increment relationship of compressive strength with different amounts of nano-SiO_2_.

When the curing age was 3 d, the compressive strength of the specimens containing 0.1%, 0.5%, and 1% nano-SiO_2_ was 0.99 MPa, 1.09 MPa, and 1.68 MPa, respectively. Compared to the specimens only reinforced with glass fiber, the compressive strength of the specimens containing 0.1% nano-SiO_2_ increased by 47%, while that containing 0.5% and 1% nano-SiO_2_ decreased by 13% and 4%, respectively. This indicates that the incorporation of a small amount of nano-SiO_2_ is adverse to the early compressive strength of the specimen. After 7 days of curing, the specimens containing 0.1% nano-SiO_2_ exhibited a maximum compressive strength, increasing to 4.38 MPa. After 14 days of curing, the compressive strength increased by 55%. After 28 days of curing, the addition of nano-SiO_2_ resulted in a more than 19% increase in compressive strength for the specimens. However, the compressive strength of the specimens containing 1% nano-SiO_2_ was actually lower than that containing 0.5% nano-SiO_2_. This may have been caused by the short curing age, large nano-SiO_2_ dosage, and incomplete reaction of nano-SiO_2_, which had a negative impact on the compressive strength of the specimens.

When nano-SiO_2_ was added, the peak compressive strength of the specimen corresponded to an increase in strain, while the strain of the specimen did not increase with the increase in the nano-SiO_2_ dosage. The maximum increase in compressive strength and strain of the specimen was 14% at the curing age of 3 d, 104% at the curing age of 7 d, 68% at the curing age of 14 d, while there was a minimum decrease of 6% at the curing age of 28 d. The results show that the compressive strength and strain of the specimen decreased after nano-SiO_2_ was added at the curing age of 28 d. Therefore, the addition of nano-SiO_2_ greatly increased the early strain of the specimen rather than the late strain, and the specimen strain was not linearly correlated with the nano-SiO_2_ dosage.

#### 3.1.3. Effect of Different Curing Ages on Compressive Strength

Figure 11 shows the relationship between different curing ages and peak compressive strength.

The compressive strength of nanocomposite fiber bonded specimens was higher than that of the control specimens at different curing ages. This indicates that the addition of nano-SiO_2_ can improve the compressive strength of the specimens to a certain extent. As the curing age increased, and the more nano-SiO_2_ added, the higher the slope and faster the growth rate of the peak compressive strength curve of the specimen in the early curing stage. When the nano-SiO_2_ dosage was within 1%, the more nano-SiO_2_ that was added, the better the early compressive strength of the specimen was. However, the later strength of the specimen did not increase with the increase in the nano-SiO_2_ dosage. According to the fitting equations of different dosages of nano-SiO_2_ and glass fiber in the figure, the fitting degree R^2^ was above 0.9626. However, for specimens with a nano-SiO_2_ dosage of 0.1%, the growth of compressive strength and the curing age had a high correlation. The fitting equation curve shows the slowdown of compressive strength at varying degrees during the curing process from 14 d to 28 d. This suggests that the compressive strength slope of the specimen changed gently within a certain curing age.

Figure 12 shows the stress–strain increment relationship of compressive strength of specimens at different curing ages.

At the curing age of 3 d, the compressive strength of the specimen containing 1% nano-SiO_2_ increased by 47% compared to that without nano-SiO_2_. At the curing age of 7 d, the compressive strength of the specimen containing 1% nano-SiO_2_ was only 3% higher than that without nano-SiO_2_. This indicates that a small amount of nano-SiO_2_ had little effect on the early compressive strength of the specimen. After 28 days of curing, the growth of compressive strength of the specimen slowed down further. Only the specimen containing 0.1% nano-SiO_2_ exhibited an increase in slope after 28 days of curing, and the compressive strength of the specimen containing 0.5% nano-SiO_2_ was higher than that containing 1% nano-SiO_2_. This could be attributed to the excessive presence of nano-SiO_2_ particles that fail to undergo a pozzolanic reaction, thereby impeding cement hydration and hindering the enhancement of compressive strength under the nano-SiO_2_ dosage of 1% [47].

After the addition of nano-SiO_2_, the strain of the specimen increased significantly at the curing age of 3 d, 7 d, and 14 d. Even after 7 days of curing, the strain of specimens containing 1% nano-SiO_2_ increased by 104% compared to those without nano-SiO_2_. At the curing age of 3 d, the strain of the specimens with different nano-SiO_2_ dosages increased by about 10%. At the curing age of 7 d and 14 d, the strain of the specimens with different nano-SiO_2_ dosages increased significantly, with an average increase of about 60% compared to that of the specimen reinforced with 0.3% glass fiber. However, at the curing age of 28 d, the strain of the specimens with different nano-SiO_2_ dosages decreased by 18%, 20%, and 6%, respectively. This is due to the fact that the added enhancement of nanoscale materials prevented the development of cracks within the composites, and the increase in the maintenance cycle allowed for better reactive bonding with other materials, which required more energy to initiate and extend the cracks and led to specimen failure [40].

### 3.2. Tensile Strength Test

#### 3.2.1. Tensile Failure Mode

Taking a specimen with a nano-SiO_2_ dosage of 0.5% as an example, the tensile failure morphology of the cemented specimens made of coal-based solid waste and nanocomposite fibers was obtained, as shown in Figure 13:(1)Under the action of axial pressure, two cracks were generated from the upper end of the specimen at the curing age of 3 d. With the increase in the load, the cracks extended downward. Ultimately, one crack passed through the entire specimen, while the other one extended to approximately one-fourth of the specimen’s length. The sample was not directly broken into two parts along the crack, and the sample still has a certain amount of residual tensile strength.(2)The cracks of the specimen at the curing age of 7 d developed from the bottom, and the cracks of the specimen extended upward under the action of axial force. As the cracks developed upward, the width of the cracks decreased gradually, and the cracks penetrated the whole lower part of the specimen, while these cracks were almost invisible in the upper part of the specimen. Compared with the specimen at the curing age of 3 d and 7 d, the cracks developed in different positions and extended in different directions. The crack width in the specimen at the curing age of 7 d was much smaller than that of the specimen at the curing age of 3 d.(3)The cracks of the specimen at the curing age of 14 d appeared from the bottom and extended upward. Although the cracks did not run through the specimen on the surface, the tensile strength dropped to a certain low value. After the specimen was removed from the press machine, the specimen did not break into two parts. The crack width of the specimen at the curing age of 14 d was smaller than that of the specimen at the curing ages of 3 d and 7 d.(4)The specimen at the curing age of 28 d had the same failure mode as that at the curing age of 14 d. That is, the cracks did not run through the specimen on the surface. The cracks of the specimen at the curing age of 28 d were generated from the bottom. With the increase in axial force, the cracks developed and extended upwards. During the extension of the cracks, an audible sound was produced.

With the increase in the curing age, the width of the cracks of the cemented specimen modified by nanocomposite fibers decreased gradually, and there was no through crack on the surface of specimens at the curing ages of 14 d and 28 d.

#### 3.2.2. Effect of Different Nano-SiO_2_ Dosages on Tensile Strength of Specimens

Figure 14 shows the tensile strength of cemented backfill materials with different nano-SiO_2_ dosages.

Figure 15 shows the increase in tensile strength of cemented specimens with different nano-SiO_2_ dosages compared to those reinforced with glass fiber only.

As shown in Figure 15, at the curing age of 3 d, 7 d, and 14 d, the tensile strength of the specimens containing 0.1% nano-SiO_2_ was lower than that of the specimens with only glass fiber added. This can be explained as follows: due to the small amount of nano-SiO_2_, nano-SiO_2_ failed to participate in the volcanic ash reaction, resulting in lower tensile strength of the specimen. At the curing age of 28 d, the specimen containing 0.1% nano-SiO_2_ had the highest slope of the stress–strain curve of tensile strength and the largest increase in the tensile strength. At the curing age of 14 d, the specimen containing 0.5% nano-SiO_2_ had the largest increasing amplitude in tensile strength. At the curing age of 7 d, the specimen containing 1% nano-SiO_2_ had the largest increasing amplitude in tensile strength. The results indicate that the addition of nano-SiO_2_ dosage had a significant impact on the increase in tensile strength of the specimens at different curing ages.

As shown in Figure 15, at the curing age of 3 d, 7 d, and 14 d, the tensile strength of cemented specimens modified by nanocomposite fiber decreased to varying degrees compared to the specimens without nano-SiO_2_. However, as the nano-SiO_2_ dosage increased, the decreasing amplitude gradually decreased. When the nano-SiO_2_ dosage was 1%, the tensile strength of the specimen increased significantly, and the highest increase in the tensile strength was 82%. When the nano-SiO_2_ dosage was 0.1%, the tensile strength of the specimen decreased by 15%. When the curing age was 28 days and the nano-SiO_2_ dosage was 0.1%, a large amount of pozzolanic reaction was produced by nano-SiO_2_, which promoted cement hydration and led to the formation of abundant C-S-H gel [47]. As a result, the tensile strength of the cemented specimens modified by nanocomposite fiber was enhanced by 47% compared to that of specimens reinforced with glass fiber only.

#### 3.2.3. Impact of Different Curing Ages on Tensile Strength of Specimens

Figure 16 shows the relationship between different curing ages and the peak tensile strength of specimens.

The relationship between tensile strength and curing age of blank control specimens and specimens with 0%, 0.1%, and 0.5% nano-SiO_2_ can be expressed by a quadratic polynomial. The fitting degree R^2^ of the two factors reaches 0.9491 or above. With the increase in curing age, the growth rate of the tensile strength of blank control specimens gradually slows down, even showing a downward trend. As the curing age increases to 28 d, the curve of tensile strength tends to flatten.

### 3.3. Shear Strength Tests

#### 3.3.1. Shear Failure Mode

Taking a specimen at the shear angle of 40° as an example, the shear failure morphology of the cemented specimens made of coal-based solid waste and nanocomposite fibers was obtained, as shown in Figure 17.

(1)With the increase in load, cracks in the specimen at the curing age of 3 d were generated along the shear angle. When the cracks developed to the bottom of the specimen, the cracks bifurcated; one crack continued to develop along the shear angle, while the other one developed to the specimen’s corners. Through the development of cracks along the shear angle, a large number of small cracks were generated on the upper part, especially at the corners of the specimen. Finally, the specimen was penetrated by the cracks, while the integrity of the specimen was maintained.(2)The specimen at the curing age of 7 d was not damaged along the shear angle. With the increase in load, a large number of small cracks were initiated in the upper part of the specimen, and a peeling phenomenon occurred; the cracks at the specimen edge expanded with the increase in load, and ultimately, the edge of the specimen fell off.(3)The cracks in the specimen at the curing age of 14 d were developed from the position of the fixture gasket (the edge of the specimen). With the increase in load, the cracks gradually expanded and had a significant effect on the specimen. The surface of the specimen began to buckle and has a tendency to fall off; the specimen edges were gradually broken, but the edges did not fall off.(4)The penetration crack of the specimen at the curing age of 28 d did not develop along the shear angle, and the two cracks penetrated the whole specimen at a certain angle with the shear angle. During the process of crack development, a loud sound was emitted from the specimen twice in succession; at the same time, the specimen was penetrated and the upper corners of the specimen were crushed, while the integrity of the specimen was still maintained, and no fragments fell off.

The shear failure morphology of cemented specimens modified by nanocomposite fibers and glass fibers was quite different. Specifically, cemented specimens modified by glass fibers were destroyed along the shear angle, while the cracks of cemented specimens modified by nanocomposite fibers had a certain angle with the shear angle, and there were a lot of small fissures generated around the main crack.

#### 3.3.2. The Effect of Different Nano-SiO_2_ Dosages on Shear Strength of Specimens

Figure 18 shows the relationship between the shear strength of specimens modified by nanocomposite fibers under various shear angles and nano-SiO_2_ dosages.

As shown in Figure 18, it is concluded that:

When the shear angle was 40°, the shear strength of the specimens modified by nanocomposite fibers was greater than that of control specimens, and the shear strength of the specimens modified by nanocomposite fibers increased with the increase in the nano-SiO_2_ dosage. When the shear angle was 50°, the shear strength of the specimen increased with the increase in nano-SiO_2_ dosage. The shear strength of the specimen containing 1% nano-SiO_2_ at the curing age of 3 d and 7 d significantly improved; the specimen containing 0.5% nano-SiO_2_ at the curing age of 14 d and 28 d exhibited the fastest increase in the shear strength. When the shear angle was 60°, the small addition of nano-SiO_2_ in the early curing stage had little effect on the shear strength of the specimen. When the curing age reached 14 d, the shear strength of the specimen showed a trend of first increasing and then decreasing with the increase in the nano-SiO_2_ dosage. When the curing time was 28 d, the shear strength of the specimen increased with the increase in the nano-SiO_2_ dosage; the specimen containing 1% nano-SiO_2_ exhibited the highest shear strength slope and the fastest increase.

Figure 19 shows the early and late shear strength of nanocomposite fiber cemented materials under various shear angles.

It can be seen from the figure that:

The early shear strength of the specimens modified by nanocomposite fibers significantly improved after the curing age of 7 d and 28 d. When the shear angle was 40°, the shear strength of the specimens containing 0.1%, 0.5%, and 1% nano-SiO_2_ increased by 25%, 55%, and 126% compared to the specimens only reinforced with glass fiber. However, when the shear angle was 40°, 50°, or 60°, the specimens modified by nanocomposite fibers had the same increasing amplitude of shear strength, and the shear strength of the specimens increased with the increase in nano-SiO_2_ dosage. At the curing age of 7 d, the specimen containing 1% nano-SiO_2_ had the largest increasing amplitude in the shear strength, increasing by 126%, 117%, and 70% under shear angles of 40°, 50°, and 60°. Therefore, the addition of more nano-SiO_2_ significantly improved the early shear strength of the specimen, and as the shear angle increased, the increasing amplitude in the shear strength of the specimen also decreased.

In the later curing stage, compared with the specimens only mixed with glass fiber, the shear strength of the specimens mixed with nano-SiO_2_ increased by more than 53%. At the curing age of 28 d, the shear strength of the specimens containing 1% nano-SiO_2_ had the largest increase, increasing by more than 130%. However, the shear strength of the specimens containing 0.1% and 0.5% nano-SiO_2_ increased by more than 50% and 70%, respectively. Therefore, the addition of nano-SiO_2_ significantly improved the shear strength of the specimens mixed with nano-SiO_2_ in the later stage, and the shear strength increased with the increase in the nano-SiO_2_ dosage.

#### 3.3.3. Influence of Curing Age on Shear Strength of Specimens

Table 5 shows the peak shear strength of specimens under various shear angles and curing ages. Figure 20 shows the incremental relationship between shear strength and a nano-SiO_2_ dosage of 0.5% at different curing ages.

As shown in Figure 20 and Figure 21, after adding 0.5% nano-SiO_2_, the shear strength of all specimens increased, except the specimens containing 0.5% nano-SiO_2_ at the curing age of 3 d under a shear angle of 60°, which decreased by 46%. When the shear angle was 40°, the shear strength of specimens at curing ages of 3 d, 7 d, 14 d, and 28 d was 0.62 MPa, 1.50 MPa, 2.26 MPa, and 2.30 MPa, respectively. In this case, the shear strength of specimens increased with the increase in curing age. At the curing age of 3 d, the shear strength of the specimen was 0.62 MPa, which was 100% higher than that of the control specimen. At the curing age of 28 d, the shear strength of the specimens increased by 92% compared to those only mixed with glass fiber. When the shear angle was 50°, the specimen presented the largest stress increase at a curing age of 14 d, and the stress increase reached 103%. When the shear angle was 60°, the specimen experienced a stress increase reaching 85% at the curing age of 28 d. As shown in Figure 20, there was a stable increasing amplitude in the shear strength of the specimen at the curing age of 28 d, of about 85% or above. However, at the curing ages of 3 d and 7 d, the addition of nano-SiO_2_ had a smaller effect on the early strength improvement of the specimen and a greater effect on the later strength improvement of the specimen.

Figure 21 shows the shear strength changes under different curing ages and shear angles.

It can be seen that the shear strength of the cemented specimen modified by nanocomposite fibers decreased with the increase in shear angle and increased with the increase in curing age [50]. When the shear angle is taken as the *x*-axis and the curing age as the *y*-axis, the shear strength surface equations of (b), (c), and (d) can be obtained through surface fitting as follows:(1)$$f_{\left(x,y\right)}=-0.04174x+0.05784y+2.619$$
(2)$$\begin{array}{l} f_{\left(x,y\right)}=8.483\times 10^{-5}y^{3}+0.0001108xy^{2}-0.001373y^{2}-0.004398xy\\ -0.01333x+0.4624y+0.3928 \end{array}$$
(3)$$\begin{array}{l} f_{\left(x,y\right)}=8.792\times 10^{-5}y^{3}+0.0001128xy^{2}-0.0002048x^{2}y-0.001396y^{2}\\ +0.00171x^{2}+0.01602xy-0.1841x-0.03151y+4.537 \end{array}$$

The sum of the squares of the residuals of each surface equation is 0.801, 0.9694, and 0.9874. The fitting equation of the largest shear strength of the cemented specimen modified by nanocomposite fibers and the curing age is obtained, and the fitting degree can be as high as 96%.

## 4. Conclusions

(1)The compressive strength of cemented materials modified by nanocomposite fibers increases with the increase in nano-SiO_2_ dosage at a curing age of 7 d. The specimen containing 1% nano-SiO_2_ has a peak compressive strength which is 84% higher than that of cemented materials only reinforced with glass fibers. When 1% nano-SiO_2_ is added, the tensile strength of the specimen at the curing age of 28 d is significantly improved, which is 82% higher than that of the cemented material only reinforced with glass fiber. When 1% nano-SiO_2_ is added, the shear strength of the specimens at the curing age of 28 d under the shear angles of 40°, 50°, and 60° reaches its peak, with a large slope and the fastest increase in shear strength. Among them, the shear strength of the cemented materials modified by nanocomposite fibers under a shear angle of 40° increases by 142% compared to those only reinforced with glass fibers.(2)When 1% nano-SiO_2_ is added to cemented material, the relationship between the growth of compressive strength and curing ages is linearly correlated with a quadratic polynomial. The tensile strength and curing age of specimens with a nano-SiO_2_ dosage of 0%, 0.1%, and 0.5% are linearly correlated with quadratic polynomials, with a fitting degree R^2^ of 0.9999 for specimens containing 1% nano-SiO_2_. The shear strength, shear angle, and curing age of cemented materials modified by nanocomposite fibers exhibit a cubic polynomial relationship, with a fitting degree of 96%.

## Figures and Tables

**Figure 1 materials-16-05314-f001:**
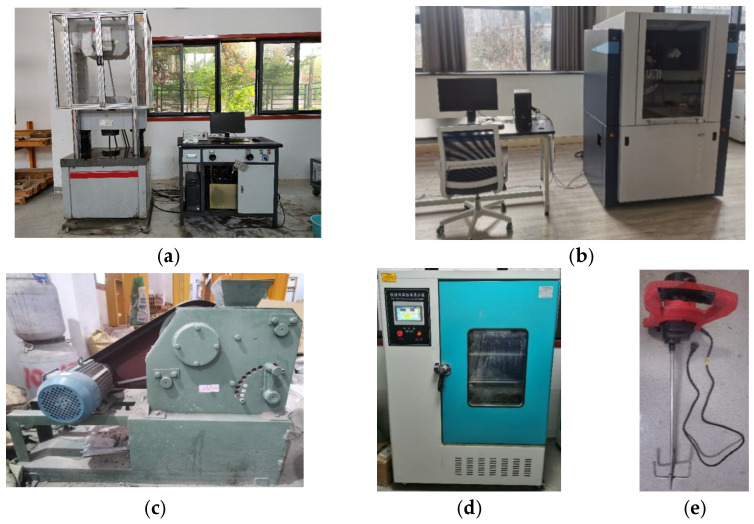
Actual picture of the test instrument: (**a**) Electro-hydraulic servo press; (**b**) XRD; (**c**) jaw crusher; (**d**) standard maintenance box; (**e**) multifunctional stirrer.

**Figure 2 materials-16-05314-f002:**
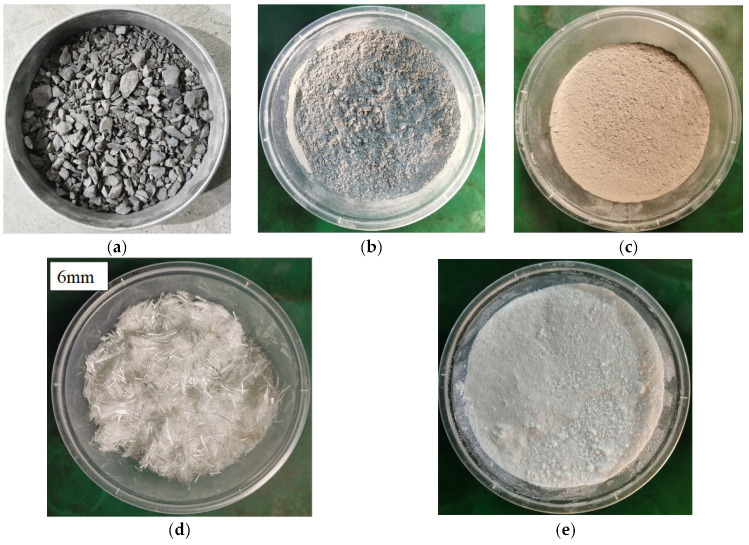
Components of cemented composites: (**a**) coal gangue; (**b**) fly ash; (**c**) Cement; (**d**) glass fiber; (**e**) nano-SiO_2_.

**Figure 3 materials-16-05314-f003:**
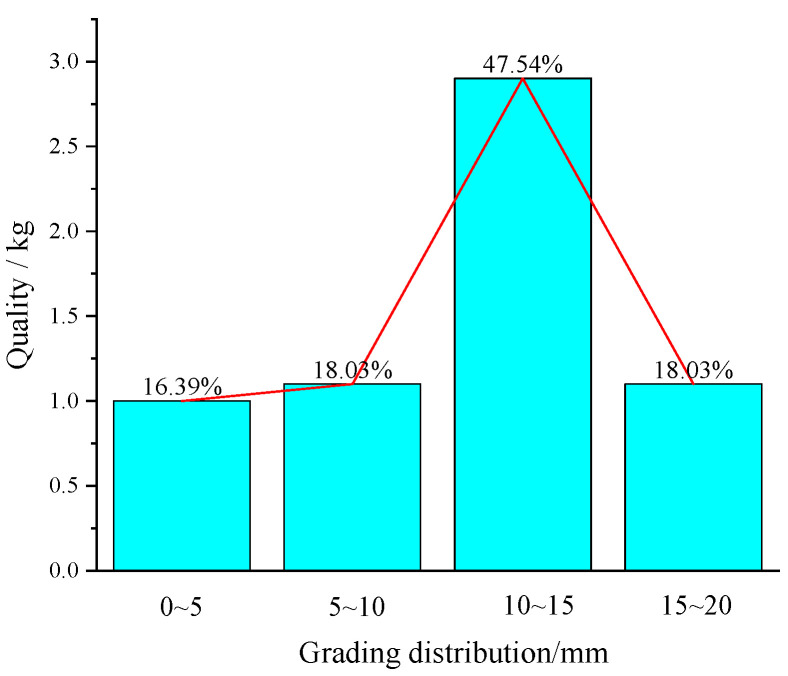
Grading distribution of gangue samples.

**Figure 4 materials-16-05314-f004:**
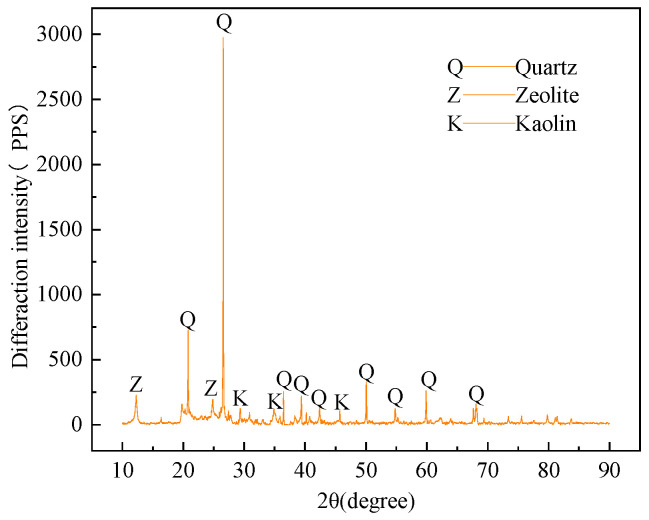
X-ray diffraction pattern of coal gangue.

**Figure 5 materials-16-05314-f005:**
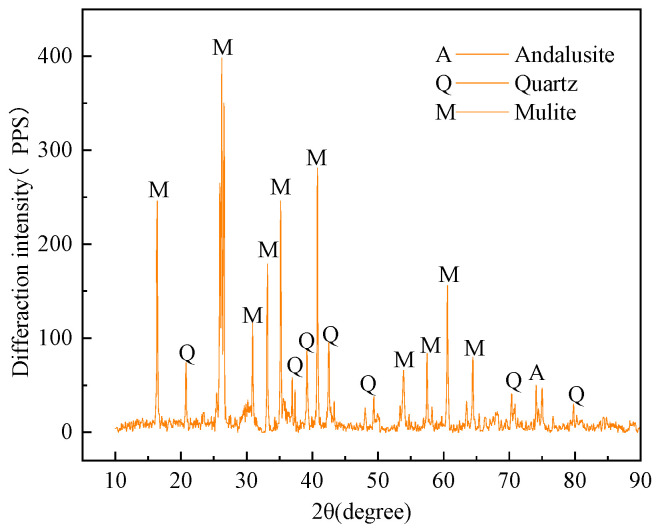
X-ray diffraction pattern of fly ash cement.

**Figure 6 materials-16-05314-f006:**
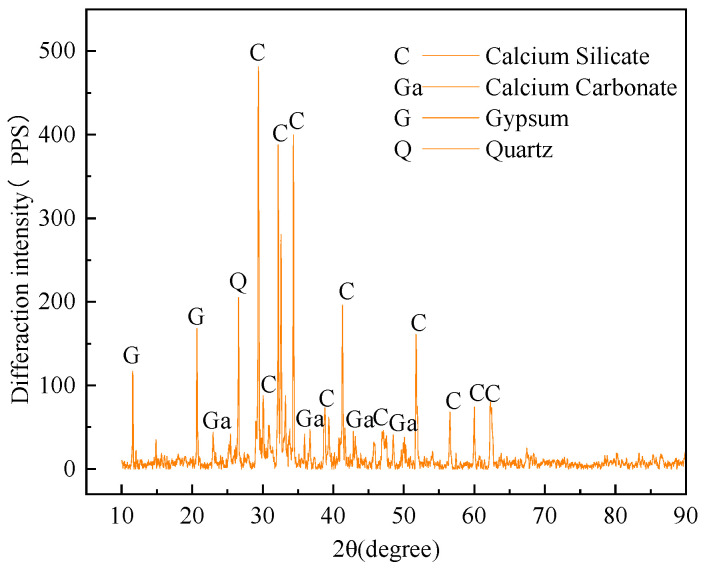
Cement X-Ray diffractogram.

**Figure 7 materials-16-05314-f007:**
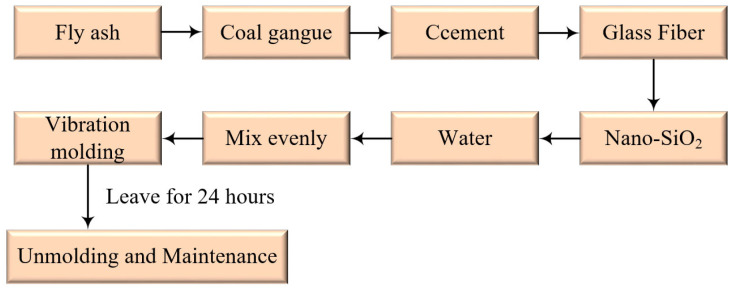
Specimen preparation process.

**Figure 8 materials-16-05314-f008:**
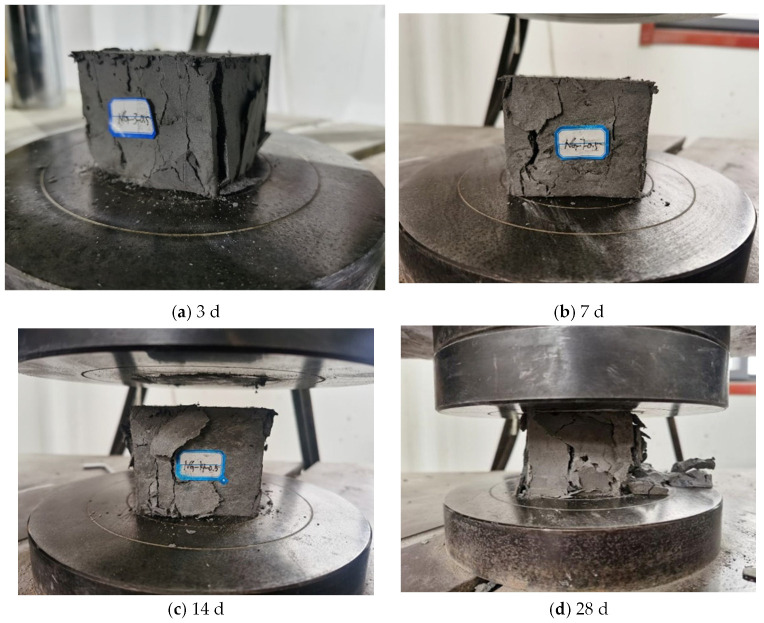
Compressive failure morphology of nanocomposite fiber specimens: (**a**) 3 d; (**b**) 7 d; (**c**) 14 d; (**d**) 28 d.

**Figure 9 materials-16-05314-f009:**
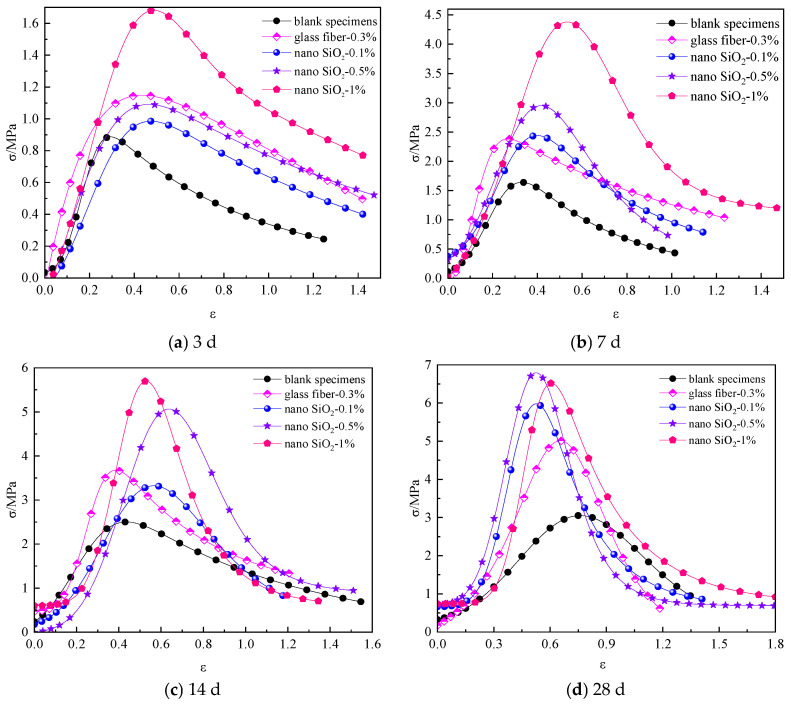
Stress–strain curves of compressive strength of cemented materials with different nano-SiO_2_ dosages: (**a**) 3 d; (**b**) 7 d; (**c**) 14 d; (**d**) 28 d.

**Figure 10 materials-16-05314-f010:**
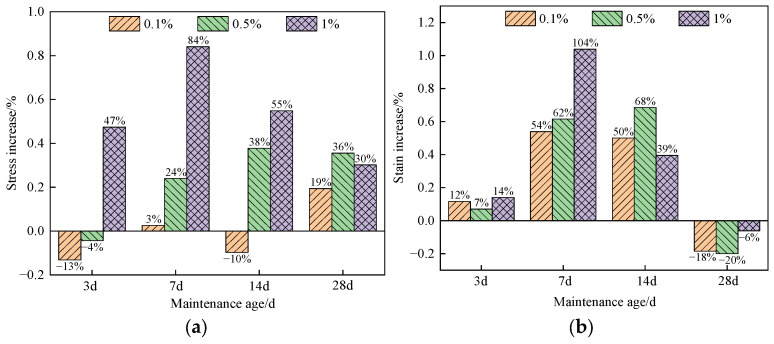
Effect of nano-SiO_2_ dosage on the stress–strain increment of the specimen: (**a**) Stress increase; (**b**) strain increase.

**Figure 11 materials-16-05314-f011:**
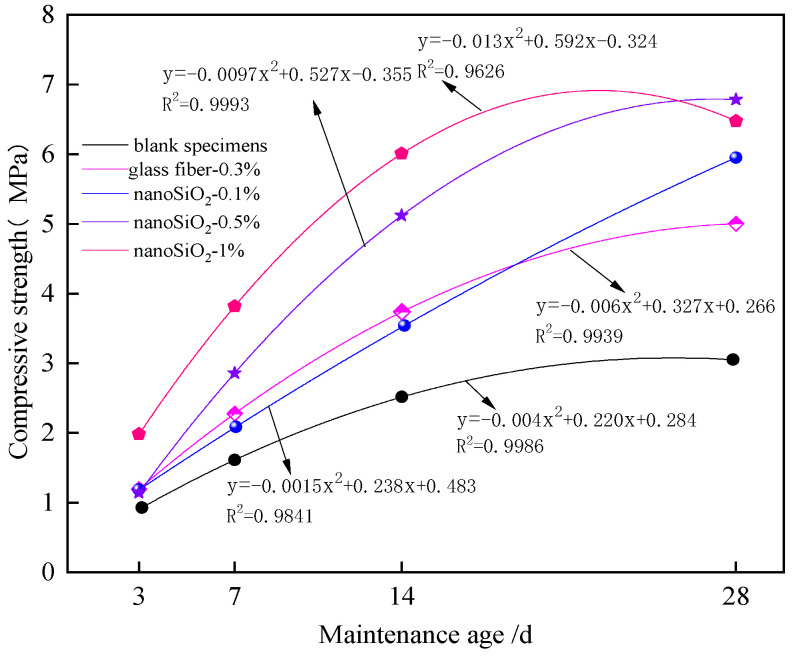
Relationship between curing age and compressive strength of specimens.

**Figure 12 materials-16-05314-f012:**
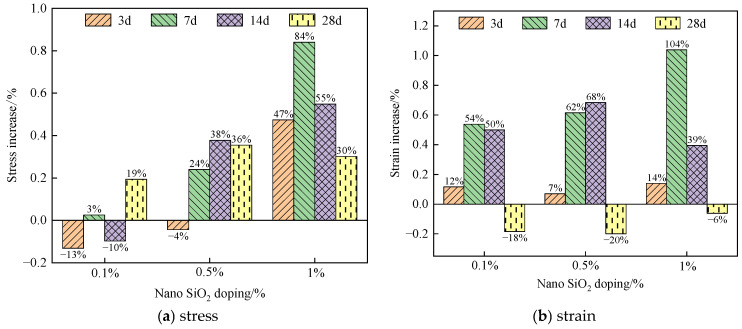
Effect of curing age on stress/strain increment of specimens: (**a**) stress; (**b**) strain.

**Figure 13 materials-16-05314-f013:**
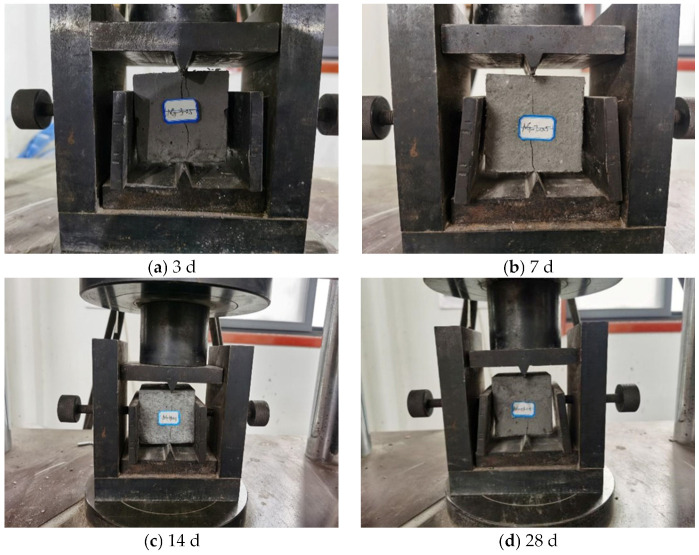
Tensile failure morphology of nanocomposite fiber Specimen: (**a**) 3 d; (**b**) 7 d; (**c**) 14 d; (**d**) 28 d.

**Figure 14 materials-16-05314-f014:**
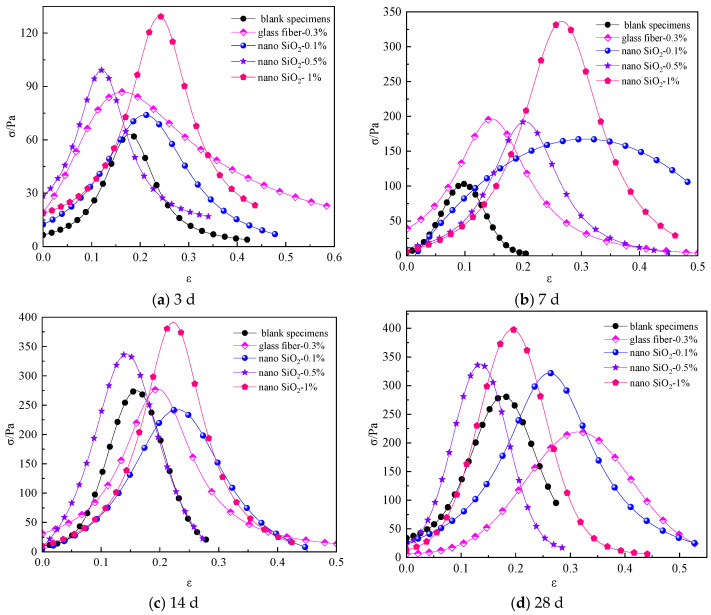
Tensile strength of specimens with different nano-SiO_2_ dosages. (**a**) 3 d. (**b**) 7 d. (**c**) 14 d. (**d**) 28 d.

**Figure 15 materials-16-05314-f015:**
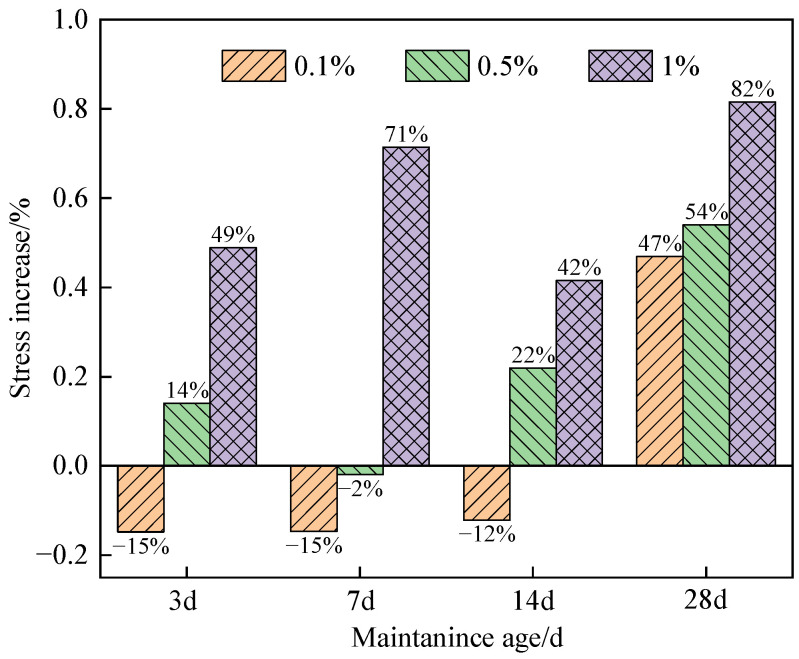
Increase in tensile strength of specimens with different nano-SiO_2_ dosages.

**Figure 16 materials-16-05314-f016:**
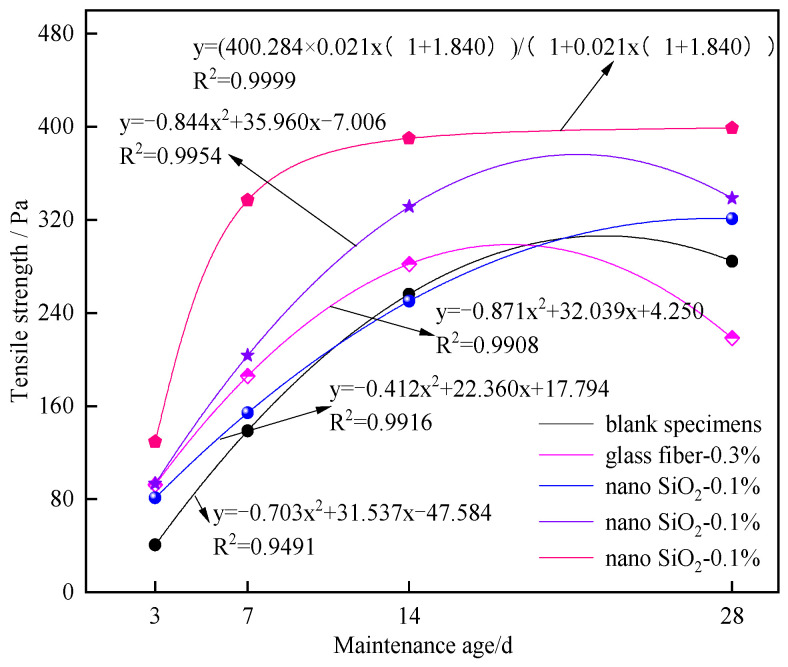
Relationship between curing age and tensile strength of specimens.

**Figure 17 materials-16-05314-f017:**
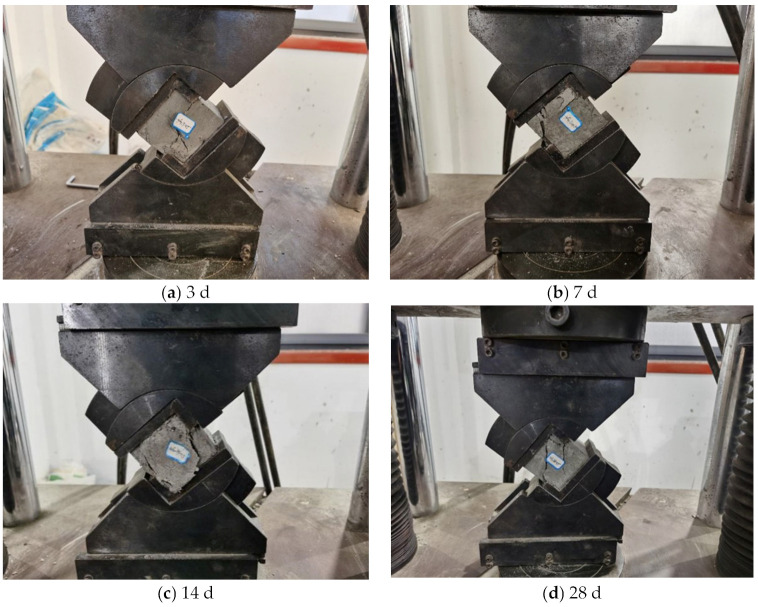
Shear failure morphology of cemented specimens modified using nanocomposite fiber at the curing age of: (**a**) 3 d; (**b**) 7 d; (**c**) 14 d; (**d**) 28 d.

**Figure 18 materials-16-05314-f018:**
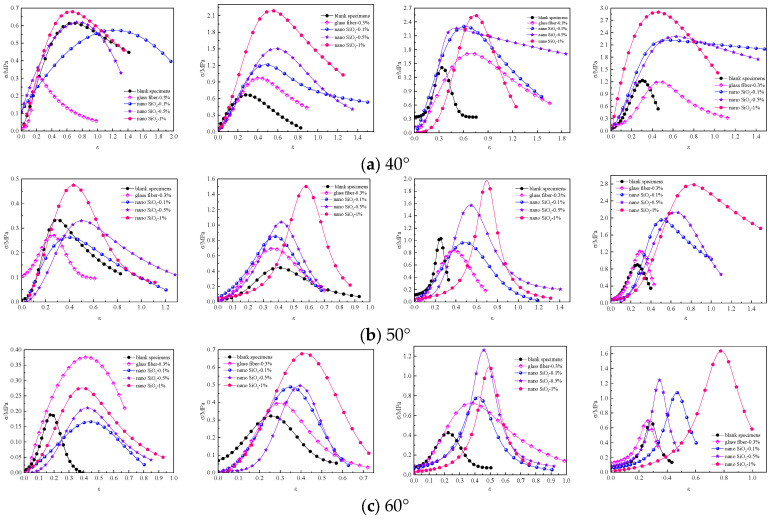
Shear strength of specimens at different shear angles: (**a**) 40°; (**b**) 50°; (**c**) 60°.

**Figure 19 materials-16-05314-f019:**
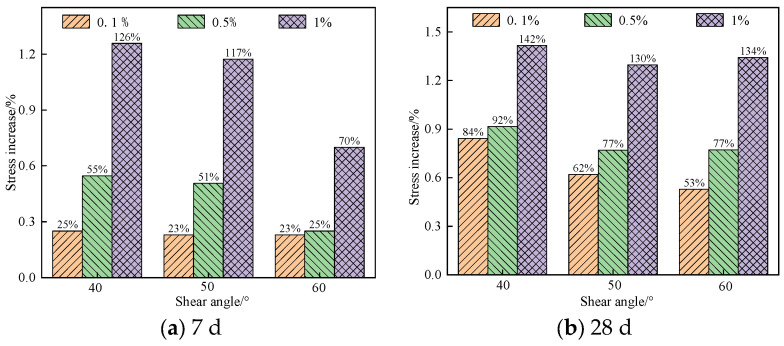
Effect of nano-SiO_2_ dosage on the increasing amplitude in shear strength of specimens: (**a**) 7 d; (**b**) 28 d.

**Figure 20 materials-16-05314-f020:**
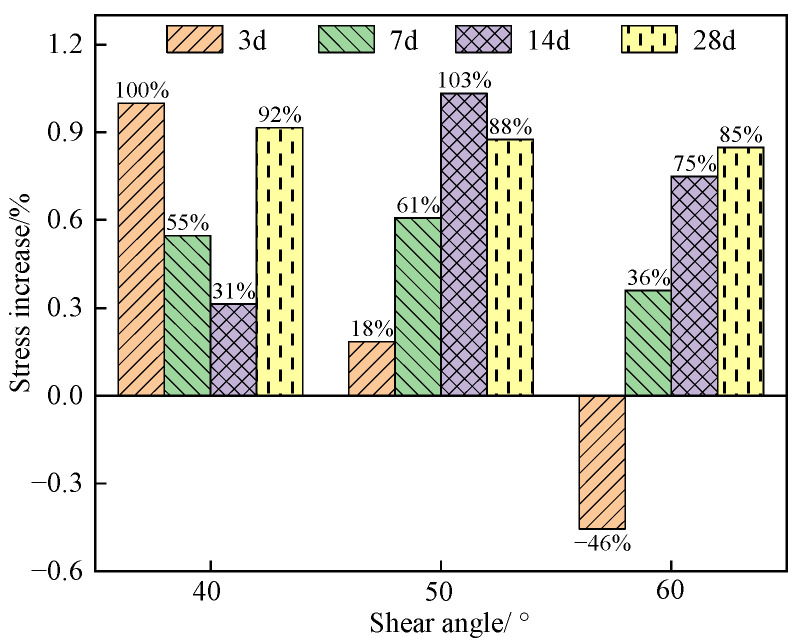
Stress increment of specimens containing 0.5% nano-SiO_2_ at different curing ages.

**Figure 21 materials-16-05314-f021:**
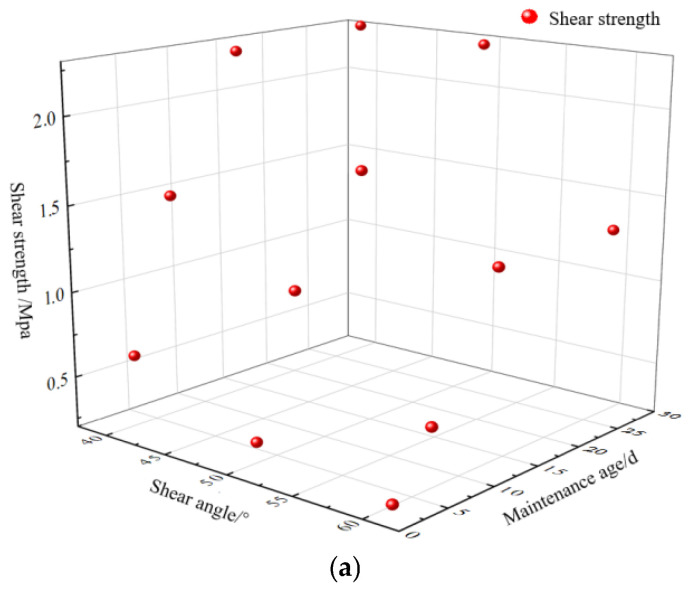
Relationship between curing age and shear strength of specimens: (**a**) 3 d; (**b**) 7 d; (**c**) 14 d; (**d**) 28 d.

**Table 1 materials-16-05314-t001:** Test equipment and equipment models.

Serial Number	Instrument/Equipment Name	Manufacturer	Model/Specification
1	Electro-hydraulic servo press	Changchun Xinte Testing machine of China (Changchun, China)	WAW-1000D
2	XRD	Dandong Tongda of China (Dandong, China)	D8ADVANCE
3	Jaw crusher	Henan Baichen Machine of China (Xingyang, China)	PE-100X125
4	Standard maintenance box	Nanjing Jinrui Testing of China (Nanjing, China)	JR-YX40
5	Multifunctional stirrer	Yongning Hongkang Trading Company of China (Hong Kong, China)	LANG-8828

**Table 2 materials-16-05314-t002:** Performance parameters of glass fibers.

Material	Non-Alkali Glass Fiber	Fiber Type	Bundled Monofilament
Specification	6 mm	Fiber density	2.699 g/cm^2^
Tensile strength	≥2000 Mpa	Elongation at break	≥2.5
Tensile modulus of elasticity	≥85 GPa	Fiber diameter	17.4 μm
Acid and alkali resistance	polar altitude	Melting point	750 °C

**Table 3 materials-16-05314-t003:** Performance parameters of nano-SiO_2_.

Material	SiO_2_	Average Particle Size	20 nm
Specific Surface Area	240 m^2^/g	Bulk density	0.06 g/cm^3^
Density	2.2~2.6 g/cm^3^	Crystal type	Ball shape
Color	White	Purity	≥99.99%

**Table 4 materials-16-05314-t004:** Test program.

Maintenance Age/d	Nano-SiO_2_ Doping	Mechanical Test			Quantities
3 d	0% 0.1% 0.5% 1%	Compressive	Tensile	Shear	60
		3	3	9	
7 d	0% 0.1% 0.5% 1%	Compressive	Tensile	Shear	60
		3	3	9	
14 d	0% 0.1% 0.5% 1%	Compressive	Tensile	Shear	60
		3	3	9	
28 d	0% 0.1% 0.5% 1%	Compressive	Tensile	Shear	60
		3	3	9	

**Table 5 materials-16-05314-t005:** Shear strength of the specimens modified by nano-SiO_2_ at different curing ages.

Maintenance Age	Shear Strength/MPa		
	40°	50°	60°
3 d	0.62	0.32	0.207
7 d	1.50	1.11	0.544
14 d	2.26	1.69	1.296
28 d	2.30	2.27	1.294

## Data Availability

The data presented in this study are available on request from the corresponding author. The data are not publicly available due to privacy.

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
