# Peer review of "Experimental Study on Mechanical Properties of Coal-Based Solid Waste Nanocomposite Fiber Cementitious Backfill Material"

_materials, 2023, doi:10.3390/ma16155314_

Round 1

Reviewer 1 Report

The paper gives a study on mechanical properties of composite material, the material propose was novel, but there are some important topics should be included

1.      Keywords were not preferred “mine solid waste” used simple one

2.      Review was little, just 23 cited reference, more recent paper should be reviewed

3.      Why did author select these temperature to curing and this humidity, is there reference or pretest to get optimal condition. .

4.      Term specimen perpetration and testing methods were lack of many information, geometry, test machines, test condition, test according to what standard

5.      How many samples used in each test

6.      All test results need deeply study comparing with other cited reference to give evidence to the readers

7.      Any mechanical testing needs fractography study optical or SEM

8.      The effect of component dependability how it was measured.

9.      How the curing age selected and why

10.   Mode of failure was needed

11.   What was the role used to control the volume fraction of each component

12.   The paper needs more data about experimental matrix with illustrated figures

Author Response

Thank you very much for your comments. Changes have been made in response to the comments, please see the attachment.

Reviewer 2 Report

The manuscript «Experimental Study on Mechanical Properties of Coal-Based Solid Waste Nano-composite Fiber Cementitious Backfill Material» by Qiangqiang Cheng, Haodong Wang, Yaben Guo, Bin Du, Qixiang Yin, Linglei Zhang, Yue Yao, Nan Zhou was submitted for peer review.
I read the submitted manuscript with great interest. The authors turned to a very urgent problem: creating a backfill based on coal waste and studying its strength characteristics after hardening.
The use of coal mining waste in backfill reduces the load on the environment by involving industrial waste into a closed production cycle.
The authors propose to use industrial waste in a material composition. The authors proposed to use a nanomodifying component to increase the strength characteristics. The manuscript addresses an interesting topic that has potential for application in mining.
Despite of the actual topic and well-conducted study, the authors have failed to prove the relevance of the study. The manuscript has significant flaws that need to be corrected. Correction of the shortcomings listed below must be done to improve the quality of the manuscript, enhance the ease of perception of the presented material and increase the interest of a readers.
1.) From my point of view, this number of keywords is very few. In addition, keywords should be more direct and related to the content of the manuscript. Keywords enable the reader to quickly search for the necessary material and enable the author to popularize their research and increase interest and citations. But if this number of keywords satisfies the requirement of the journal, this comment is advisory.
2.) The abstract is not quite formed correctly. It is very blurry and framed incorrectly. It seems that the authors have taken certain phrases from the text and thus formed the abstract. The abstract should clearly indicate the purpose of the study, its importance for society (i.e. to characterize the problem), identify the methods and materials of the study, and the conclusions should be clearly and briefly formulated. There is no "starting point" in the abstract, that is, information about previous studies (one sentence is enough). From my point of view, in the abstract, such information begins with the statement: "Previously conducted studies have established that ...".
2.1) It is desirable to avoid narrative text in the abstract.
2.2) Try to use words and phrases: an analysis has been carried out; studied; developed; proposed; established and so on. It is advisable to start sentences in the abstract with these words and phrases.
2.3) At the end of the abstract, it is necessary to indicate the result obtained by the authors, for example: A model has been developed that allows ...; A dependence has been established which is...; A pattern has been revealed...; An efficient system (technology) has been proposed, and so on.
The abstract should be revised.
3.) The manuscript has a weak list of references (24 references in total). There is no full coverage of research in terms of geography of citations. There are not enough references to international studies in the field, especially to the work of Eastern European, Ukrainian, or Russian scientists. There are only 5 references to international studies, which is very few.
The list of references is intended to demonstrate the depth of the authors’ study of the material, the relevance and interest of their research.
3.1.) The depth of study is demonstrated with the number of references - is not sufficient.
3.2.) Relevance – with the availability of research in recent years – is not sufficient.
3.3.) Interest – with the availability of research by scientists from different countries - is not sufficient (practically absent).
I ask the authors to take this recommendation seriously. Since you are publishing your manuscript in an international publication, it is necessary to demonstrate the international relevance and interest of this issue. This can be done by analyzing the studies of scientists from different countries. It is imperative to supplement the list of references with studies of scientists from different countries over the past 3-5 years to show geographical (general/global) interest and relevance.
Major revision of References might be sufficient if these tests have been performed. Otherwise, the paper should be considered as rejected in the present form.
Below I present a few papers relevant to this study that could greatly improve the manuscript. The authors have the right to use the material proposed or offer their own versions of international studies to increase the geography of citation.
The list of references must be completed.
4.) In the introduction when analyzing previous studies, the authors make inaccuracies or provide information that overloads the text and often their claims are not accompanied with evidence. It is important for readers to know the essence (main idea) of the research you are referring to when analyzing previous work.
In the introduction, it is necessary to analyze the previously completed work and note what has been done, what are the shortcomings, and what has been done incorrectly.
Such flaws are present throughout the introduction. The authors need to revise the introduction, make adjustments, and supplement their statements with evidence.
4.1) I am not a native speaker, but nevertheless, in my opining, the authors form a very long sentences, which are very difficult to perceive. Such sentences greatly reduce the easy perception of the material.
4.2) In the introduction, the authors refer to several works and quite rightly state what is done in this study. However, the authors do not explain why this study is interesting: what has been done right or wrong, what can be learned from the study, what needs to be corrected or improved and why this research is important.
4.3) The authors make very long statements and there is no proof (references) for these statements.
5.) From my point of view, the authors abuse the names of scientists when mentioning the study, for example Peng Zhang et al. and so on. A reference [16] is sufficient. This can be avoided, as the authors have done in lines 36, 41, 55, 58. If the reader is interested in the name of the researcher, then it is easy to refer to the references list. It is important for the reader to know the essence (main idea) of the disclosed issue, not the name of the researcher.
5.) I would recommend avoiding group references, for example [1-6]. From my point of view, allowed up to three; more than three references are not acceptable and must be deciphered. Each paper you refer is unique and the studies you refer deserve more proper and careful review to demonstrate (and prove) its importance for the current research. It is necessary to demonstrate in detail the essence of each study and their need for your work. It has already been noted in recommendation (4.2) that you have many statements without indicating awareness. You will avoid group references by correcting this fact.
6.) At the end of the introduction, there is no brief conclusion of the analytical study of earlier papers. The authors did not summarize their analysis and did not identify unresolved issues. This conclusion should make it possible to characterize the actual question posed, the purpose of the study and the tasks to be solved to achieve this goal. For example: Analyzing the above, it can be noted that ... is a very topical issue. Therefore, the purpose of this study is ... and to achieve this, it is necessary to solve the following tasks: 1); 2); ... Such a conclusion allows the reader to understand the vector of the study, and the authors to correctly formulate the conclusions. It needs to be improved.
7.) Considering the comments (3) and (4) I would like to note that the authors have very poorly disclosed the main subject of the study.
The impact of waste on the environment is great, so the issues of reducing this impact are very relevant and scientists around the world are trying to minimise it. In recent years, a lot of work has been carried out on the study of a material created on the basis of enrichment tailings with the aim of their subsequent use in a closed, waste-free (low-waste) production or for products intended for civil engineering.
For example,
7.1) Ermolovich, E.A.; Ivannikov, A.L.; Khayrutdinov, M.M.; Kongar-Syuryun, C.B.; Tyulyaeva, Y.S. Creation of a Nanomodified Backfill Based on the Waste from Enrichment of Water-Soluble Ores. Materials 2022, 15(10), 3689. https://doi.org/10.3390/ma15103689.
This work is similar in terms of goals, objectives and research methods to the manuscript submitted for peer review. The authors study a composite based on waste from the processing of water-soluble ores. To increase the strength characteristics of the created material, fullerene-astarlene is used as a nanomodified additive. From my point of view, this work should be used in the analysis of previously performed studies, since it uses the methods of mechanical, microstructural, X-ray phase and petrographic analyzes to confirm arguments.
7.2) Kongar-Syuryun, Ch.B.; Faradzhov, V.V.; Tyulyaeva, Yu.S.; Khayrutdinov, A.M. Effect of activating treatment of halite flotation waste in backfill mixture preparation. Mining Informational and Analytical Bulletin 2021, 2021(1), 43–57. https://doi.org/10.25018/0236-1493-2021-1-0-43-57.
7.3) Khayrutdinov, A.; Kongar-Syuryun, Ch.; Kowalik, T.; Faradzhov, V. Improvement of the backfilling characteristics by activation of halite enrichment waste for non-waste geotechnology. IOP Conf. Ser.: Mater. Sci. Eng. 2020, 867(1), 012018. https://doi.org/10.1088/1757-899X/867/1/012018.
In studies (7.2), (7.3), the authors propose the activation treatment of tailings before mixing to improve the strength and rheological characteristics. Activation treatment of the components or the addition of some kind of activating additive is one of the ways to improve the quality of the created material.
From my point of view, the works (7.3) and (7.4) will suit the authors in the analysis of previously completed works to demonstrate various options for controlling the characteristics of the created composite.
If the authors become familiar with the works presented in (7.1), (7.2), (7.3) they will be able to properly form the introduction, enrich their manuscript with international research by scientists from Poland, Czech Republic, Slovenia, Slovakia, Russia, Germany and demonstrate the depth of their material, as well as eliminate the remarks (3) and (4).
8) Of particular interest to me, and I think readers as well, is composite and its composition. Readers need to know all this so that they can repeat the experiment.
9.) When describing an experiment to create a backfill, you must specify:
9.1) how was the convergence of the results achieved;
9.2) how was the homogenization (mixing) carried out; what is the mixing tool; what is the mixing velocity and time;
9.3) what is the sequence of filling of the components;
9.4) it is also not clear how the homogeneity of the composition (thoroughness of mixing) was achieved, provided that the amount of some components (nanomodifying additive) in the composite is minimal;
9.5) what equipment was used to study samples for uniaxial compression;
9.6) how many samples were prepared;
To eliminate remarks (8) – (9), I would recommend reading the work (7.1). The recommended paper is similar to the one submitted for peer review. The article (7.2) describes the methodology in sufficient details.
Summary: The manuscript is not a finished research work. Corrections are needed. The chosen research topic is relevant. From my point of view, the authors failed to present their research correctly and clearly, which reduced its value and worsened the ease of perception of the material presented.
From my point of view, the manuscript cannot be published in the open press without correction in accordance with my suggestions.

Author Response

(The authors gave the same response as above.)

Round 2

Reviewer 1 Report

accepted in the present form

Reviewer 2 Report

The manuscript "Experimental Study on Mechanical Properties of Coal-Based Solid Waste Nano-composite Fiber Cementitious Backfill Material" by Qiangqiang Cheng, Haodong Wang, Yaben Guo, Bin Du, Qixiang Yin, Linglei Zhang, Yue Yao, Nan Zhou was submitted for second review.

As can be seen from the submitted manuscript and the explanatory note to the review, the authors did a lot of work to make changes in accordance with the comments. The revised manuscript is a completed scientific study on a highly relevant topic. The revised version of the manuscript, in my opinion, fully satisfies the requirements of a scientific article and can be published in the open press.